

# Grazer density and songbird counts in a restored conservation area

Lilla Lovász[1,2], Fränzi Korner-Nievergelt[3] and Valentin Amrhein[1,2]

[1] Research Station Petite Camargue Alsacienne, Saint-Louis, France
[2] Department of Environmental Sciences, Zoology, Universität Basel, Basel, Switzerland
[3] Oikostat GmbH, Ettiswil, Switzerland

## ABSTRACT

Grazing by large herbivores is increasingly used as a management tool in European nature reserves. The aim is usually to support an open but heterogeneous habitat and its corresponding plant and animal communities. Previous studies showed that birds may profit from grazing but that the effect varies among bird species. Such studies often compared bird counts among grazed areas with different stocking rates of herbivores. Here, we investigated how space use of Konik horses and Highland cattle is related to bird counts in a recently restored conservation area with a year-round natural grazing management. We equipped five horses and five cattle with GPS collars and correlated the density of their GPS positions on the grazed area with the density of bird observations from winter through the breeding season. We found that in the songbirds of our study site, both the overall density of bird individuals and the number of species increased with increasing density of GPS positions of grazers. Correlations of bird density with horse density were similar to correlations with cattle density. Of the eight most common songbird species observed in our study area, the Eurasian Skylark and the Common Starling had the clearest positive correlations with grazer density, while the Blackbird showed a negative correlation. Skylarks and Starlings in our study area thus seem to profit from year-round natural grazing by a mixed group of horses and cattle.

## INTRODUCTION

Bird communities in open landscapes are often positively influenced by ungulate grazing, due to the heterogenous, structure-rich environment created by the grazers (*Roth, 1976*; *Van Klink et al., 2016*; *VanWieren, 1995*; *Vera, 2000*). Although a negative influence of grazing on birds is sometimes reported, for example, due to intense grazing on farmland (*Dross et al., 2018*), low-intensity grazing is generally agreed to be beneficial for most bird species, especially those of higher conservation concern (*Nikolov, 2010*). Therefore, grazing by one or more species of large herbivores is increasingly used as a management tool in European nature conservation areas (*Henning et al., 2017*; *Loucougaray, Bonis & Bouzillé, 2004*; *Rosenthal, Schrautzer & Eichberg, 2012*).

The extent to which bird species react to grazing likely depends on how much they rely on the particular niches affected by grazing (*Milchunas, Sala & Lauenroth, 1988*).

Corresponding author
Lilla Lovász, lilla.lovasz@unibas.ch

For example, shortened vegetation may provide suitable nesting habitat and higher food availability and accessibility for some bird species (*Leal et al., 2019*; *Toepfer & Stubbe, 2001*), while others may be impeded by the effect of trampling (*Sharps et al., 2017*). The Eurasian Skylark *Alauda arvensis* is one example of a species that was shown to require open and structurally diverse habitat mosaics with relatively short vegetation to maximize the number of nesting attempts (*Toepfer & Stubbe, 2001*; *Wilson et al., 1997*). While the Skylark seems to generally profit from grazing, trampling was reported to be a main cause of nest loss on meadows grazed by livestock at high densities (*Pavel, 2004*).

Studies so far mainly compared how the impact of grazing on breeding bird communities differs between enclosures with different stocking rates of large herbivores (*Báldi, Batáry & Erdős, 2005*; *Dross et al., 2018*). For example, *Batáry, Báldi & Erdős (2007)* found that grassland birds were more abundant on extensively grazed areas compared to intensively grazed areas, while this was not the case in non-grassland birds.

However, the habitat use by grazers within an enclosure is usually not homogenous (*Gander et al., 2003*; *King & Gurnell, 2005*). For understanding the influence of space use patterns of grazers on birds within a given grazed area, it may help to obtain position data of individual grazers. One example of such a study is *Köhler, Hiller & Tischew (2016)*, who investigated space use of horses in relation to a bird assemblage in a German nature reserve by using a GPS collar on one horse. The authors found that the density of bird observations, especially in the Skylark, was higher where the density of horse GPS positions was higher.

Here, we studied how counts of songbirds from winter through the breeding season are related to the space use of a mixed assemblage of five Konik horses and five Highland cattle in a French nature reserve that was recently ecologically restored. In this study area, the applied management approach is natural grazing, a low intensity (<0.5 animal units per hectare) year-round mixed grazing regime with the aim of substituting extinct wild herbivores such as the wild horse (*Equus ferus*) or the aurochs (*Bos primigenius*) with domestic breeds kept in semi-wild conditions, that is, without systematic winter feeding and with minimal human intervention (*Linnell et al., 2015*; *Vermeulen, 2015*). We investigated how the overall counts of songbird individuals and of the number of songbird species correlated with the density of grazer GPS positions, and how the correlations varied among songbird species and in horses vs cattle.

## MATERIALS AND METHODS

### Study site

Our study site (Fig. 1) is located on the Rhine island of the nature reserve Petite Camargue Alsacienne in France, north of Basel, Switzerland. About 100 ha of the island has been part of an ecosystem restoration project since 2014. During the restoration process, the former crop fields on the area have been turned into an alluvial environment. A mixed habitat of grassland scattered with bushes (hawthorn, dog rose) and gravel sites was constructed, surrounded by patches of old forests (oak, ash). Since the beginning of the restoration project, saplings of willow and poplar are increasingly growing on some parts

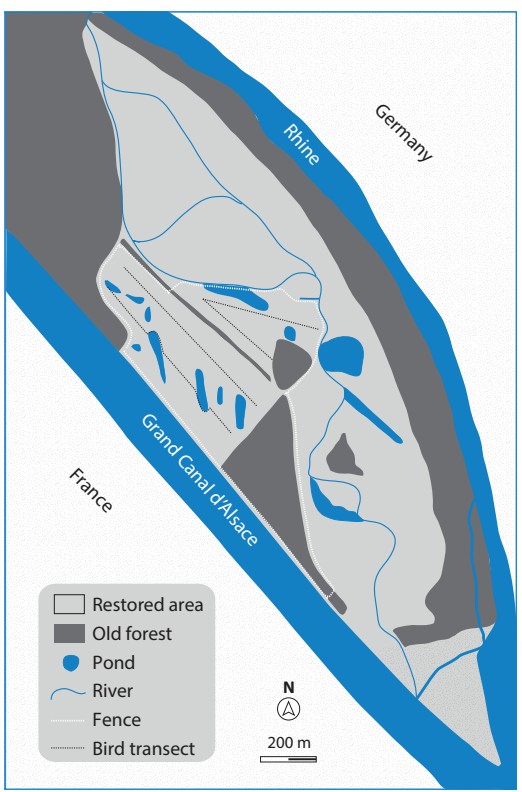

**Figure 1 Map of the study site.** The ecologically restored area (approximately 100 ha) on the Rhine island of the nature reserve Petite Camargue Alsacienne in France, and the 32-ha study site (test grazing-area), marked by the white dashed line. 

of the area. The water of the Rhine is led through the island in small creeks, and several ground-water ponds have been created.

The study was done with permission of the national nature reserve Petite Camargue Alsacienne.

## Grazer data

Konik horses and Highland cattle were gradually introduced into a 32-ha test enclosure on the island between September 2018 and March 2019 to contribute to the maintenance of the heterogenous and open habitat. We equipped all horses ($n = 5$) and cattle ($n = 5$) with GPS collars (Followit, type Pellego) recording their positions once per hour, starting from the time of their arrival to the area. We used data starting from January 2019 when three cows and all five horses were present on the area; two additional oxen arrived in March 2019. The overall grazing pressure thus was approximately 0.3 animal per hectare. The data were downloaded through satellite processing from the interface of the GPS collar provider (Followit, Lindesberg, Sweden), therefore no contact to the animals was necessary to access the data. Since decades, GPS collars have been widely used on cattle without causing harm or disturbance (*Turner et al., 2000*; *Ungar et al., 2005*), and as recently discussed by *Collins et al. (2014)*, GPS collars also comply with animal welfare requirements for horses.

GPS accuracy may be affected by atmospheric conditions, satellite or receiver errors (*Hurn, 1993*), satellite geometry (*Dussault et al., 2001*), topography, overhead canopies, or adjacent structures (*Di Orio, Callas & Schaefer, 2003*; *Moen et al., 1996*); therefore the GPS fixes in our dataset likely had some imprecision. Our applied GPS collars did not record HDOP (horizontal dilution of precision) data and we therefore did not correct for inaccuracy of the fixes. However, since only 3.32% of all grazer positions fell outside the fenced area (those fixes were not included into the analysis), we assumed that this rate would not strongly influence our results (*Ganskopp & Johnson, 2007*). We considered the hourly GPS positions of the grazers as describing their "space use" (i.e., the density distribution of horses and cattle over the study area).

## Bird data

In 2019, we made 22 bird surveys between 31 January and 24 July. Visits were carried out in favorable weather conditions, on days without rain and with little or no wind.

We surveyed bird abundance by transect walking on the grazed meadows; we did not include a 10.6-ha-area of old forests that was part of the enclosure, so that the final size of the studied area was 21.4 ha. We selected three line transects (*Gregory, Gibbons & Donald, 2004*; *Laiolo, 2005*) over the meadow area, each of about 700 m length, so that all parts of the grazed meadow were in visual and/or auditory distance from a transect. A trained observer (L.L.) walked along the transects with a slow pace and marked the position of the observed birds on a digital map (Map Marker 2.11_1442). During each survey, all identified individuals from all bird species were recorded; this was our response variable "bird counts". Birds flying higher than 20 m above the ground without showing connection to the area were excluded (e.g., Skylarks that made territorial songflights at >20 m elevation were counted, but raptors crossing >20 m over the meadows or migrating Common Swifts *Apus apus* were not). Surveys were conducted in the mornings until noon, avoiding dawn hours to minimize detectability differences due to rapid changes in the birds' conspicuousness and activity (*Dawson, 1981*). The order of visits of the three transects per morning were alternated systematically. Differences in bird detectability between transects were probably rather small, due to the similar open habitat of the surveyed areas. To minimize the risk of double counts, we used a cut-off distance of 60 m to either side of the transects so that transects would not overlap but cover the entire grazed area, and followed the recommendation of *Dawson & Bull (1975)*: unless it is reasonably sure that the same individual is observed, observations are counted as different individuals.

## Statistical analysis

For analysis, the study area was divided into 113 50 × 50 m grid cells using the corner points of the UTM grid. Our measure of bird density was the number of bird counts per survey per grid cell. Horse and cattle GPS positions (taken once per hour) were summed up per grid cell over the last 30 days prior to a bird survey, resulting in our measure of grazer (horse and cattle) density. We assumed that grazer space use patterns earlier than 30 days before the respective bird survey did not substantially influence space use of
birds. Because we assumed detectability of bird species to be relatively homogeneous across the study area and our aim was not estimating the total number of birds present in the study area, we did not take detectability into account in our analyses (*Buckland et al., 2001*). Due to the migratory behavior of some bird species, species composition changed over the course of the study. For the analyses, we excluded periods when a migratory bird species was not regularly present in our study site, which was from the 9th survey session (17th April) for the Pipits; until the 7th survey session (23rd March) for the Barn Swallow; and until the 12th survey session (4th May) for the Red-backed Shrike. The surveys were distributed between winter, spring and summer in order to capture a large variety of environmental conditions (e.g., temperature, vegetation) as well as different bird behaviors (wintering, migrating, breeding).

We used a negative binomial mixed model with a logarithm link function to measure species-specific correlations between bird counts and grazer density. The logarithm of the size of grid cells was used as an offset in the linear predictor in order to make counts comparable between grid cells of different sizes (at the edges of the study area, some parts of the grid cells fell outside of the fenced area). We log-transformed grazer densities and therefore replaced values of zero (i.e., zero observations in a grid cell) with half of the minimum non-zero value (*Bellégo & Pape, 2019*). The log-transformed grazer density was used as covariate, and bird species was included as a random factor. Both random intercepts and random slopes were used to model species-specific correlations between bird and grazer density.

We fitted the model using Bayesian methods as implemented in Stan (*Stan Development Team, 2014*) via the function brm from the package brms (*Bürkner, 2017*) in R 3.6.1 (*R Core Team, 2016*). The default flat prior distributions over the reals were used for the average correlation between bird and grazer density. Half-student $t(3,0,10)$ was used for the variance parameters, and Gamma (0.01, 0.01) was used as prior distribution for the shape parameter of the negative binomial distribution.

We assessed model fit by residual analyses and posterior predictive model checking. From the residuals we calculated a semi-variogram to check for spatial correlation, and we calculated the autocorrelations to check for temporal correlation. The semi-variance ranged between 2.5 and 3.5 over the distances 0–200 m and it did not increase with distance. Temporal autocorrelations measured within species and within the $50 \times 50$ m grid cell ranged from −0.002 to 0.004 for the lag of 1–10 weeks, and thus we judged these temporal correlations to be small enough to be ignored. We further simulated 2,000 different virtual replicated data sets from the model (posterior predictive distribution) and compared the proportion of zero values as well as the variance between the replicated and the real data to check for zero-inflation and overdispersion. The proportion of zero values in the replicated data ranged from 0.97 to 0.98 (1% and 99% quantiles), which included the proportion of zero values in the data (0.98). Also, the standard deviation of the data (1.89) fell within the range of standard deviations of the replicated data from the model (0.80 to 8.21). Therefore, we concluded that the model described the variance and the proportion of counts of zero of our data well and did not suffer from apparent spatial and temporal correlation.

We used 2,000 simulated random values from the joint posterior distribution of the model parameters to describe parameter estimates and their uncertainty. We used the median of the marginal posterior distribution as point estimate and the 2.5% and 97.5% quantiles as lower and upper limits of the 95% Bayesian compatibility intervals (*Amrhein, Greenland & McShane, 2019*; *Amrhein, Trafimow & Greenland, 2019*).

Code and data are available on an online repository: https://osf.io/g8a6t/?view_only=86a5e3a5f3b54d7aa7681519e4b7df39.

## Declaration of analysis and reporting decisions

This is an exploratory study (*Amrhein, Korner-Nievergelt & Roth, 2017*) describing observations of birds and positions of grazers in our study site. Before starting data collection, we did not know which bird species would have sufficient sample sizes for analysis; we selected the eight most suitable study species after looking at the data. In the revision of the paper, as suggested by the referees, we added an analysis on species-specific correlations of horse vs cattle densities with bird count density that we did early in our study but had not reported in the first version of the paper, and we added two new analyses on the correlations between overall songbird density and grazer density and overall songbird species richness and grazer density.

## RESULTS

In total, we observed 2,125 individuals from 64 bird species, among them 1,620 individuals from 34 songbird species (order Passeriformes; Table 1). The eight most common species that had sufficient sample sizes for statistical analysis ($n > 20$ counts) and were clearly connected to the grazed area of our study site all belonged to the songbirds (Table 1). From those eight species, we made a total of 1,424 observations. The only species that certainly bred on the grazed area were the Skylark and the Red-Backed Shrike (these birds were observed showing territorial behavior such as songflights, or breeding behavior such as feeding chicks); the White Wagtail may have bred in the study area as well (the habitat was suitable but we did not observe signs of breeding). The Great Tit, Common Starling and Common Blackbird bred in the bushes and patches of forest in and around the fenced area. Barn Swallows were observed foraging in flight and Pipits (Water Pipit *Anthus spinoletta* or Meadow Pipit *Anthus pratensis*) mainly in flocks on the ground.

Median grazer density (numbers of GPS positions per grid cell for the last 30 days prior to a bird survey) did not change markedly over the course of the study (Fig. 2). Variance in grazer density increased in May, indicating that grazing occurred homogeneously on all cells in winter, while during spring and summer some cells were grazed with a higher intensity whereas others were largely avoided by the grazers.

As a first analysis, we fitted our model on correlations between bird count density and grazer density by using separate predictor variables for the densities of horse and cattle positions (thus correcting each grazer species effect for the other grazer species). The correlation between the densities of horse and cattle positions was $r = 0.35$. Figure 3 shows that model predictions for the effects of horse vs cattle densities on bird

**Table 1 List of all observed birds.**

| Songbirds | Nr of individuals | Other birds | Nr of individuals |
|---|---|---|---|
| **Common Starling** *Sturnus vulgaris* | 505 | Common Swift *Apus apus* | 123 |
| **Barn Swallow** *Hirundo rustica* | 347 | Mallard *Anas platyrhynchos* | 72 |
| **Eurasian Skylark** *Alauda arvensis* | 215 | Mute Swan *Cygnus olor* | 70 |
| **Pipits** *Anthus sp.* | 194 | Tufted Duck *Aythya fuligula* | 56 |
| **Great Tit** *Parus major* | 68 | Eurasian Teal *Anas crecca* | 33 |
| **Common Blackbird** *Turdus merula* | 37 | Green Sandpiper *Tringa ochropus* | 31 |
| **Red-backed Shrike** *Lanius collurio* | 32 | Grey Heron *Ardea cinerea* | 29 |
| **White Wagtail** *Motacilla alba* | 26 | Eurasian Coot *Fulica atra* | 22 |
| Common House Martin *Delichon urbicum* | 41 | Common Kingfisher *Alcedo atthis* | 6 |
| Winter Wren *Troglodytes troglodytes* | 20 | Red-crested Pochard *Netta rufina* | 6 |
| Carrion Crow *Corvus corone* | 17 | Ruddy Shelduck *Tadorna ferruginea* | 6 |
| Long-tailed Tit *Aegithalos caudatus* | 15 | Common Buzzard *Buteo buteo* | 5 |
| Reed Bunting *Emberiza schoeniclus* | 13 | White Stork *Ciconia ciconia* | 5 |
| European Robin *Erithacus rubecula* | 12 | Little Grebe *Tachybaptus ruficollis* | 5 |
| Blue Tit *Cyanistes caeruleus* | 9 | Wood Sandpiper *Tringa glareola* | 5 |
| Fieldfare *Turdus pilaris* | 8 | Common Redshank *Tringa totanus* | 5 |
| Common Raven *Corvus corax* | 7 | Little Ringed Plover *Charadrius dubius* | 4 |
| Common Chaffinch *Fringilla coelebs* | 7 | Black Kite *Milvus migrans* | 4 |
| European Greenfinch *Carduelis chloris* | 6 | Common Snipe *Gallinago gallinago* | 3 |
| Common Chiffchaff *Phylloscopus collybita* | 6 | Northern Shoveler *Spatula clypeata* | 3 |
| Garden Warbler *Sylvia borin* | 5 | Little Egret *Egretta garzetta* | 2 |
| Yellowhammer *Emberiza citrinella* | 4 | Eurasian Wryneck *Jynx torquilla* | 2 |
| Grey Wagtail *Motacilla cinerea* | 4 | Ferruginous Duck *Aythya nyroca* | 1 |

(Continued)

| Songbirds | Nr of individuals | Other birds | Nr of individuals |
|---|---|---|---|
| Eurasian Blackcap *Sylvia atricapilla* | 4 | Middle Spotted Woodpecker *Leiopicus medius* | 1 |
| Common Whitethroat *Sylvia communis* | 4 | Lesser Spotted Woodpecker *Dryobates minor* | 1 |
| Eurasian Siskin *Spinus spinus* | 3 | Great Egret *Ardea alba* | 1 |
| Eurasian Reed Warbler *Acrocephalus scirpaceus* | 2 | Eurasian Hobby *Falco subbuteo* | 1 |
| European Goldfinch *Carduelis carduelis* | 2 | Ruff *Philomachus pugnax* | 1 |
| Western Yellow Wagtail *Motacilla flava* | 2 | Northern Lapwing *Vanellus vanellus* | 1 |
| Tree Pipit *Anthus trivialis* | 1 | Eurasian Sparrowhawk *Accipiter nisus* | 1 |
| Eurasian Jay *Garrulus glandarius* | 1 | | |
| Savi's warbler *Locustella luscinioides* | 1 | | |
| Dunnock *Prunella modularis* | 1 | | |
| Whinchat *Saxicola rubetra* | 1 | | |

**Note:**
The eight more closely investigated songbird species are in bold.

count densities were rather similar in the eight investigated songbird species; for all further analyses, we thus pooled the data for horse and cattle GPS positions.

Bird species that showed a relatively clear positive correlation with pooled grazer density ($P(\beta > 0)$ is relatively high; Table 2; Fig. 4), given our statistical model, were Starlings and Skylarks. In the Starling, our data are most compatible with slopes between 0.28 and 1.02 and in the Skylark with slopes between −0.18 and 0.63 (Table 2). Species with the clearest negative correlations were Blackbirds and Barn Swallows ($P(\beta > 0)$ is relatively low): the data on the Blackbird are most compatible with slopes between −0.92 and 0.07, and in the Barn Swallow with slopes between −1.33 and 0.29 (Table 2). Apart from the Starling, however, the patterns are quite uncertain, given the wide compatibility intervals (Table 2, Fig. 4).

In an additional analysis, we considered the overall densities of songbirds and of grazers, that is, the numbers of individuals of all songbirds and of all grazer positions summed per grid cell over the entire study period. Figure 5 shows the positive correlation between overall songbird and grazer density (model coefficient: 0.21, 95% CI [0.04–0.38]). Further, the species richness (numbers of species) of songbirds per grid cell was positively correlated with the overall grazer density (Fig. 6; model coefficient: 0.12, 95% CI [0.01–0.23]).

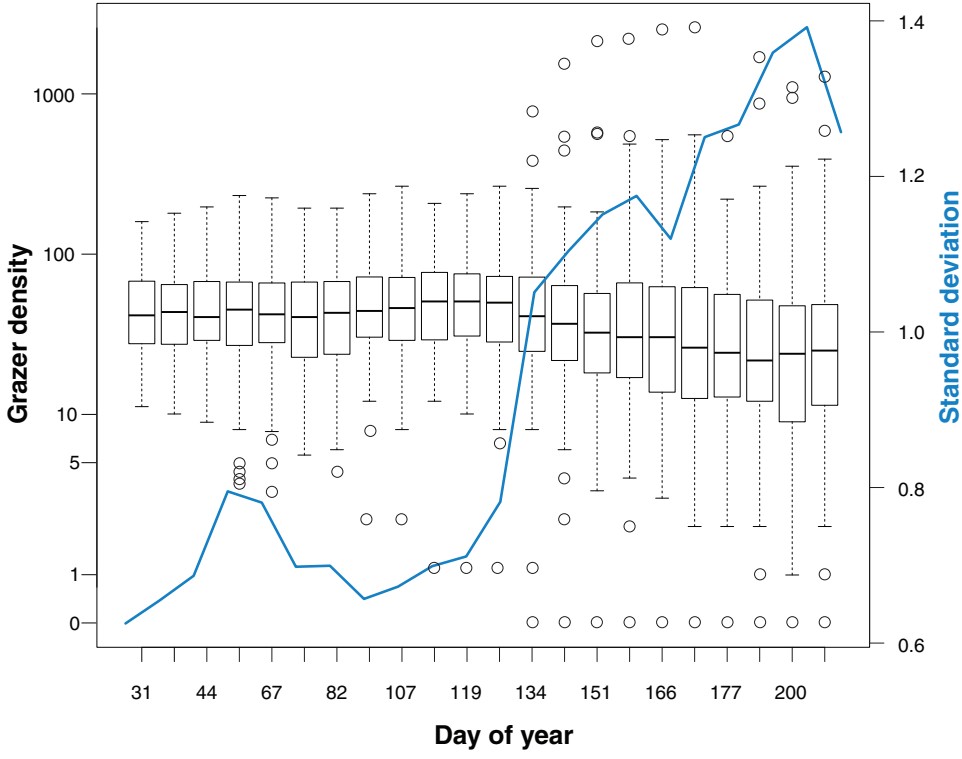

**Figure 2 Grazer density for 22 bird surveys in the course of the study period (January to July).** Boxplots show the distribution of grazer densities, given as numbers of pooled GPS positions of horses and cattle per grid cell ($n = 113$) for the last 30 days prior to a bird survey. Day of year corresponds to the dates of bird surveys (1 = 1st January). The blue line indicates the standard deviation of grazer densities for each survey.

## DISCUSSION

We investigated responses of birds to natural grazing in a newly restored nature conservation area by using GPS collars on individual cattle and horses. We studied grazing pressure on a continuous scale of grazer density, which differs from earlier studies categorizing grazing pressure on entire meadows as, for example, "high" or "low" (*Batáry, Báldi & Erdős, 2007*). Our approach takes into account that cattle and horses are known for their heterogenous habitat use (*Lamoot, Meert & Hoffmann, 2005*) and thus that a possible effect of grazing may vary within a given study site. Further, unlike previous studies that investigated either the breeding or winter season (*Hartel et al., 2014*; *Leal et al., 2019*), we considered bird observations starting from winter through the breeding season, with year-round presence of semi-wild grazers. The resulting correlations therefore not only describe density of breeding birds but average relationships between bird and grazer densities over many different environmental conditions and life-cycle stages of birds.

 We found that in the songbirds of our study site, both the overall density of individual birds and the number of species increased with increasing density of grazer positions. Among the eight most commonly observed songbird species, the density of Starling observations showed the clearest positive correlation with density of grazer positions.

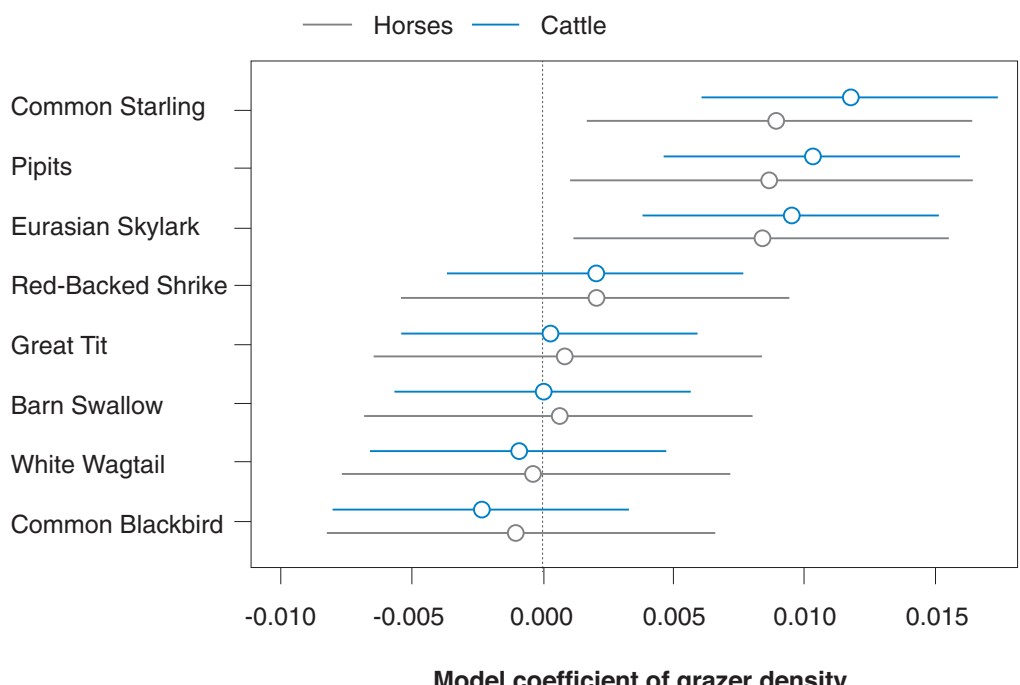

**Figure 3** Species-specific correlations of horse and cattle density (numbers of GPS positions per grid cell for the last 30 days prior to a bird survey) with bird count density (numbers of bird counts per survey per grid cell) for the eight most common songbird species. Given are medians (circles) and 95% Bayesian compatibility intervals (lines) of the posterior distributions of the fitted values.

**Table 2** Characteristics of the marginal posterior distributions of the model parameters: medians, 2.5% and 97.5% quantiles (limits of the 95% Bayesian compatibility interval) and proportions of posterior mass above zero ($P(\beta > 0)$). The posterior mass corresponds to the posterior probability of the hypothesis that the parameter value is positive; values close to 1 indicate strong evidence for a positive relationship, values close to zero indicate strong evidence for a negative relationship.

| Parameter | Median of posterior | 2.5% quantile | 97.5% quantile | $P(\beta > 0)$ |
|---|---|---|---|---|
| Intercept | −10.5 | −11.7 | −9.2 | – |
| Grazer density average | 0.02 | −0.44 | 0.48 | 0.53 |
| Grazer density Starling | 0.62 | 0.28 | 1.02 | >0.99 |
| Grazer density Skylark | 0.21 | −0.18 | 0.63 | 0.86 |
| Grazer density Red-backed Shrike | 0.13 | −0.35 | 0.67 | 0.71 |
| Grazer density Pipits | 0.12 | −0.60 | 0.86 | 0.65 |
| Grazer density Great Tit | 0.03 | −0.40 | 0.45 | 0.56 |
| Grazer density Wagtail | −0.16 | −0.74 | 0.38 | 0.26 |
| Grazer density Blackbird | −0.38 | −0.92 | 0.07 | 0.06 |
| Grazer density Barn Swallow | −0.48 | −1.33 | 0.29 | 0.12 |
| SD species intercept | 1.47 | 0.86 | 3.13 | – |
| SD species grazer density | 0.51 | 0.21 | 1.22 | – |
| Negative binomial shape | 0.013 | 0.011 | 0.015 | – |

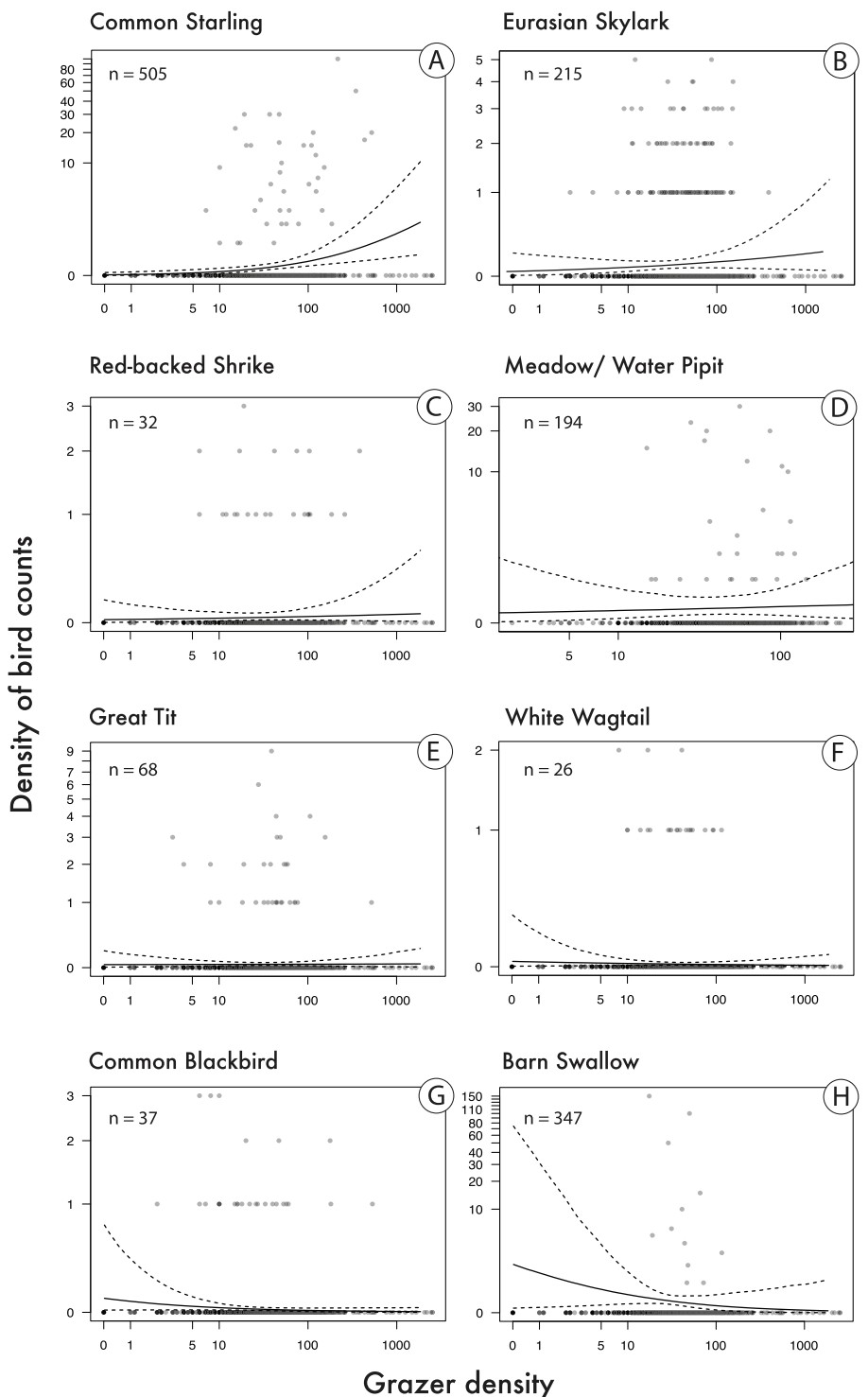

**Figure 4 Correlations between bird count density (numbers of bird counts per survey per grid cell) and grazer density (numbers of pooled GPS positions of horses and cattle per grid cell for the last 30 days prior to a bird survey).** Given are medians (solid lines) and 95% Bayesian compatibility intervals (dotted lines) of the posterior distributions of the fitted values for eight songbird species (A–H). Sample sizes (*n*) refer to the total number of birds counted in 113 grid cells during 22 surveys.

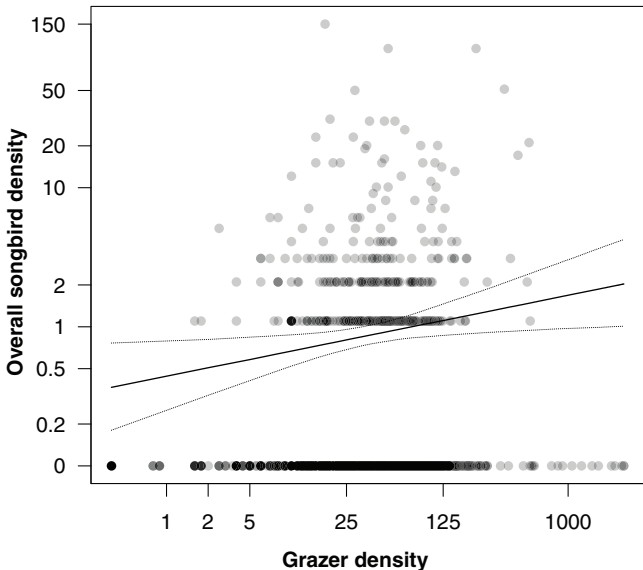

**Figure 5 Correlation between overall songbird density (numbers of individuals of all songbird species per survey per grid cell) and grazer density (numbers of pooled GPS positions of horses and cattle per grid cell for the last 30 days prior to a bird survey).** Given are medians (solid line) and 95% Bayesian compatibility interval (dotted lines) of the posterior distributions of the fitted values.

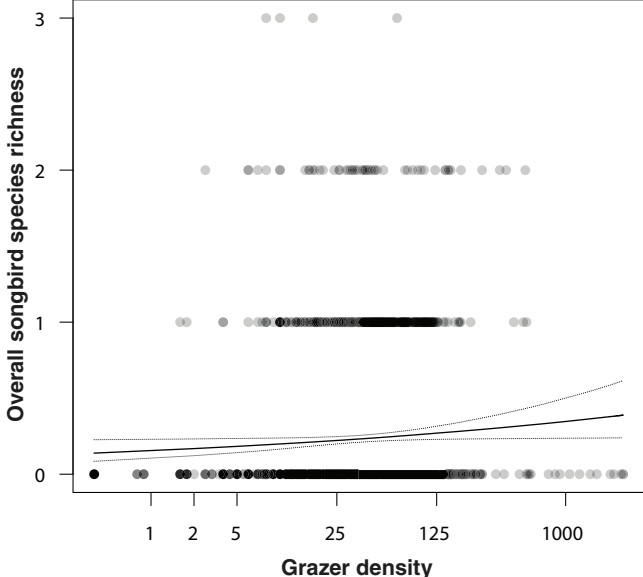

**Figure 6 Correlation between overall songbird species richness (numbers of all songbird species per survey per grid cell) and grazer density (numbers of pooled GPS positions of horses and cattle per grid cell for the last 30 days prior to a bird survey).** Given are medians (solid line) and 95% Bayesian compatibility interval (dotted lines) of the posterior distributions of the fitted values.

This was to be expected, given that Starlings usually prefer grazed pastures rather than arable farmlands (*Heldbjerg et al., 2017*) and often follow grazing herds, profiting from flushed insects (*Källander, 2004*). We also found a relatively clear positive correlation in

the Skylark (although also slight negative correlations would be compatible with our data, given our model; *Amrhein, Greenland & McShane, 2019*; *Amrhein, Trafimow & Greenland, 2019*). Skylarks have been suggested to both benefit from and be impeded by grazing (reviewed by *Donald (2010)*). This is because trampling by large herbivores may destroy nests (*Pavel, 2004*; *Popotnik & Giuliano, 2000*), while the shortened vegetation height benefits Skylarks in terms of food availability, accessibility, and suitable nesting habitat (*Odderskær et al., 1997*; *Wilson et al., 1997*). When we looked at the density of GPS positions of horses and cattle separately, accounting for the presence of the other grazer species, we found that the correlations with bird densities were rather similar.

Our results do not necessarily imply a causal relationship between grazing and density of birds; for example, non-causal correlations between grazers and birds could arise because both prefer the same habitat. In our study site, however, the habitat was completely restored and ecological succession started from bare gravel soil in 2014. Although in the meantime some of the growing saplings were removed manually, horses and cattle contribute to keeping the vegetation short and to re-creating pioneer habitats with bare soil (e.g., at resting areas of the grazers) since autumn 2018. Although the degree of causality is hard to quantify, we think it is probably correct to say that Starlings and Skylarks seem to profit from the presence of horses and cattle by using habitats that are kept open by the grazers.

We observed the clearest negative correlations in Blackbirds and Barn Swallows. Possible explanations may be that Blackbirds are often found next to areas with more dense vegetation that may not be preferred by grazers, while Barn Swallows were often observed flying over the water ponds that naturally had low or zero densities of grazer GPS positions. The uncertainty in the correlations found for Pipits, Red-Backed Shrikes, White Wagtails and Great Tits seems too high to allow interpretation, although the slightly positive correlations in Red-Backed Shrikes and Pipits would fit what we would expect given that those species are often found on or next to areas with bare ground.

It will be interesting to investigate in future studies how the space use of birds and grazers varies depending on season and how this affects the correlations between bird and grazer densities. It would also be interesting to study the influence of vegetation and ecological succession on spatial behavior of grazers and birds, although here again it would be difficult to disentangle cause and effect. Future research could also investigate how food abundance and availability may affect the space use of birds through the indirect effect of grazing on the vegetation.

Similar to our study, *Köhler, Hiller & Tischew (2016)* and *Kerekes & Végvári (2016)* found that associations between bird abundance and grazing intensity varies greatly among bird species. Also *Neilly & Schwarzkopf (2019)* described that responses of birds to grazing are often complex and will reflect habitat requirements of the individual bird species. Whether a possible effect of natural grazing in a nature reserve is meeting conservation goals thus depends on which species one aims to protect. Among the eight most often observed birds in our study, the two species that are most threatened are the

Skylark and the Red-Backed Shrike (according to the IUCN Red List; *BirdLife, 2018*; *BirdLife International, 2017*). The observed positive correlations with grazer densities in those species are encouraging from a conservational point of view, given that natural grazing with horses and cattle is usually implemented to enhance habitat diversity and to support species of conservation concern.

## ACKNOWLEDGEMENTS

We thank the team of the Réserve Naturelle Petite Camargue Alsacienne to have made it possible to conduct our research in the nature reserve. Lisa Malm and an anonymous referee gave helpful comments on the first version of the paper.

### Funding

This work was supported by the Fondation de bienfaisance Jeanne Lovioz, the Foundation Emilia Guggenheim-Schnurr, the Ornithologische Gesellschaft Basel, the Swiss Association Pro Petite Camargue Alsacienne, the Foundation Wolfermann-Nägeli, the Foundation Frey-Clavel, and the MAVA Foundation. The funders had no role in study design, data collection and analysis, decision to publish, or preparation of the manuscript.

### Grant Disclosures

The following grant information was disclosed by the authors:
Fondation de bienfaisance Jeanne Lovioz.
Foundation Emilia Guggenheim-Schnurr.
Ornithologische Gesellschaft Basel.
Swiss Association Pro Petite Camargue Alsacienne.
Foundation Wolfermann-Nägeli.
Foundation Frey-Clavel.
MAVA Foundation.

### Competing Interests

Fränzi Korner-Nievergelt is employed by oikostat GmbH.

### Author Contributions

- Lilla Lovász conceived and designed the experiments, performed the experiments, analyzed the data, prepared figures and/or tables, authored or reviewed drafts of the paper, and approved the final draft.
- Fränzi Korner-Nievergelt conceived and designed the experiments, analyzed the data, prepared figures and/or tables, authored or reviewed drafts of the paper, and approved the final draft.
- Valentin Amrhein conceived and designed the experiments, authored or reviewed drafts of the paper, and approved the final draft.
## Animal Ethics

The following information was supplied relating to ethical approvals (i.e., approving body and any reference numbers):

Our work was a purely observational study and no ethical approval was needed. The GPS collars that provide our grazer data are used by the nature reserve (where our study site is located) for localizing their animals and we are using their data only. Fitting the collars for management purposes did not require an approval for the reserve, therefore we cannot present such approval.

## Data Availability

Data and code are available at OSF:

Lovász, L., Korner, F., & Amrhein, V. (2020). Grazer density and songbird counts. OSF. Data and code. DOI: osf.io/g8a6t.

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
