# Peer review of "Grazer density and songbird counts in a restored conservation area"

_PeerJ, doi:10.7717/peerj.10657_

## Round 0.1 · original submission · Major Revisions

We have now received two in depth reviews (see below). Both reviewers have raised some important points. Please respond to their comments in a revised version of your manuscript.

·

Basic reporting

The manuscript is well written and quickly gets to the point of the study, there are only two minor suggestions on the background information and writing:

Lines 36-38: A more balanced introduction could mention when negative effects of grazing have been observed on bird communities (e.g. in cases of over-grazing) or be more specific to what types of grazing that may be beneficial to bird communities.

Line 52: “For example, (Batary et al. 2007) found that...” should read “For example, Batary et al. (2007) found that...”.

Experimental design

The study is carried out in a systematic and purposeful way. My main point here is that the analysis is very limited in relation to the data that has been collected and the questions being asked. Please explain why the response variables were limited to counts of each species across the whole season in the manuscript, or consider expanding the analysis according to the suggestions below. Another important point is how the purpose of the study and its method is described, see comments below.

Firstly, it would also be useful to know what time of the year the birds benefit from presence of grazers, and differences to proximity of horses or cattle. There is an emphasis on the strength in carrying out the study for a long time on lines 219-224. Although this is a strength, the opportunity in describing when the birds are most dependent on grazer presence is missed. The manuscript would be strengthened by giving further detail to how the grazer-bird relationship changes over time. This would, for example, provide useful knowledge to nature reserve managers where grazing is part of the conservation measures during parts of or the whole year. (See also the comment regarding lines 253-255 below.)

Moreover, it would also be interesting to know how the overall abundance of birds and species richness of songbirds was related to grazer presence. Given the data that has been collected, these variables should be easy to incorporate in the analysis or analyse separately. If there is a reason for not including these measures in the analysis, this reason should be stated in the manuscript.

Lastly, if all grazers are marked with individual GPS collars, it should be possible to test whether the birds preferred grazing by one grazer more than the other (i.e. horses or cattle). This information is, for example, useful when selecting the type of livestock/grazers for conservation management.

Lines 253-255: This sentence makes the reader wonder why this was not investigated in the current study as the data seem to be available. I’m referring to one point in the reviewing criteria: “Coherent bodies of work should not be inappropriately subdivided merely to increase publication count.” If more data is needed, please state (briefly) what more information that is required to carry out this analysis. I also suggest making this statement more relevant to the reader by encouraging more studies on this topic from researchers in general, rather than discussing what the authors can do in the future.

Lines: 55-59: It is not clearly stated what new information the correlation of grazer GPS points and bird counts bring, in comparison to the more common method described above (measuring bird density/community under a specific herbivore density). This study would have a higher relevance to conservation management if the reason for the method used was clearer, especially when the differences in study methods is used to motivate the study. For example, is there a problem with the more common method, does the method used here provide a more robust understanding of how dependent different bird species are on recent grazing by livestock, or does the method used provide results that can confirm results obtained with another method? Moreover, adding a sentence or two after the first sentence in this paragraph (i.e. on line 56) about the ecological/conservation management implications of the knowledge obtained from this study would significantly improve this section.

Lines 218-219: Again, what new information does this approach bring to the field? If using the difference in methods as a justification for carrying out the study, it needs to be clear why another method is important.

Lines 101-102: It would be useful, if possible, to say how accurate/precise the GPS points are. Can the actual position of the livestock be assumed to fall within a certain range of the reported position? This would help in interpreting the reliability of the results.

Lines 195-196: “The only species that certainly bred on the 196 grazed area were the Skylark and the Red-Backed Shrike.“ - How was this confirmed, by observations of nests and/or displays of territorial behavior such as singing? For the species where breeding status within the survey area was uncertain, which observations were done? Providing these details would back up the statements about breeding status.

Validity of the findings

Lines 255-257: I commend the authors for being clear with the limitations in demonstrating causal effects from correlational studies. By also providing an example of the kind of study that would be required to demonstrate a causal effect, (e.g. a randomised and replicated grazer experiment) and encouraging such studies to be carried out by others, this statement gets more meaning and is more useful to readers in the same research field.

Reviewer 2 ·

Basic reporting

no comment

Experimental design

no comment

Validity of the findings

no comment

Additional comments

This paper deals with the important issue of the effects of grazing on the abundance of birds, in this example in a grassland habitat. Although other studies have already addressed this issue, this experiment brings the novelty of using grazers marked with GPS allowing for a continuous analysis of grazing pressure. Although I think this study has interesting information, I have some questions and comments that I list below, expecting those can help improve the paper.

Introduction

Regarding the introduction I think that, although much information is given, there are some aspects that were not mention and could help to frame the work. Here are some examples:

Lines 42-44: the consequences of shorter vegetation can relate not only with food availability but also to food accessibility

Line 44: It is said that other birds “may be impeded by the effect of trampling”, but why? What are the effects of trampling that can affect birds?
Although some examples are given, I miss a deeper discussion here about the direct and indirect consequences of grazing for birds, namely through the effects on vegetation height and composition, soil compaction, etc...

Line 50: The authors focused mainly on the consequences for birds during the breeding season, but grazing can have further consequences in other parts of the year. I suggest this is mentioned here.

Lines 55-56: This sentence is difficult to read

Line 60: By counts you mean abundance? Total number of individuals?

Lines 60-69: I think that this paragraph needs a revision because it is a mixture of introduction with methods (for example it is not necessary to say here how many horses and cows were marked). Maybe it would be better if you start by a small description of management/grazing regime in your study area, then explain why is your work important and what is the novelty of your study and finally present the objectives

Line 66-69: This last sentence is very similar to the 1st in this paragraph. Also, I suggest that you improved the description of your objectives, maybe by using hypotheses. What were you expecting to obtain?

Materials & Methods

Lines 76-77: Is there any reference that you can add here? I suggest you rewrite the sentences that follows so it is clear that those were actions taken as part of that restoration project

Line 92: I think it is more correct to say cows or bulls...

Line 99: Do you know what is the estimated error (in meters)?

Line 117: Is it possible to include a map of the study area? I think it will help the reader to get a better notion of the study area, habitats, transects used for data collection, etc.

Line 119: What was the distance between parallel transects?

Line 122: I understand why you considered skylarks in song flights in your analysis, but depending on the scale used to make an association between grazers and birds use of the area, it may be difficult to make a strong connection...

Line 133: Again, here by count you mean total of observations during the sampling period?

Lines 144-146: I think this needs to be better explained. As far as I understood you had hourly data for the position of the animal, for 24h. So as a measure of grazing pressure you used the sum of animals present in the cell, every hour, for 30 days prior to bird count?

Line 161: Can you please provide a Reference to sustain these criteria?

Results

Lines 194-195: I suggest adding a table in Supp Material with a list of the total species detected and number of individuals/species

Lines 197-98: Did you collect data to confirm this information? If not, this is not a result of you study but an observation (although I understand that this information will be probably important to discuss some of your findings)

Lines 200-201: Please also include information in terms of nr of grazers/ha so these results are comparable with those of other studies in terms of grazing pressure.

Line 201: In figure 1 I think it would help if you could had information about months or the separation/limits of the three seasons that you considered in the analysis (instead of just having in xx axis the number of days from 1st January)

Discussion

Lines 217-218: I think this paper has good data to add information about the effects of grazing in bird abundance, however I think that it could go further and gain extra importance and strength if the author had data regarding other variables that could help explain the patterns found (namely regarding food abundance and/or availability; changes in vegetation caused by grazing etc.). As far as I understood this data are not available, so I suggest that at least these aspects are mentioned in the Discussion.

Lines 225-236: In this paragraph you present some possible explanations for the patterns found in terms of relationship of birds with grazing intensity. One think that I feel important to know is if those patterns for each species response were maintained thorough all sampling period, or was it more evident in some of the seasons?

I did not understand if you tested the possible influence of season, but, as you mentioned, the fact that you collected data in more than one season is one relevant aspect of you work.

Line 237: I agree that this work has the novelty of analysing this effect on a continuous scale but, on the other hand, it seems to me that this sampling design makes it more difficult to attribute a causal-effect relationship between grazers density and bird abundance.

Line 245-252: Don´t you think that some of these patterns detected for species as White Wagtail and Blackbird can be influenced by the low number of birds detected?

Finally, I suggest a careful revision to all the text since I detect some typos errors.

---

## Round 0.2 · accepted · Accept

Thank you very much for making the requested changes in your revised manuscript. One reviewer and myself are happy with the changes you have made and recommend that your manuscript be accepted as is.

·

Basic reporting

No comment

Experimental design

No comment

Validity of the findings

No comment

Additional comments

I appreciate that the authors have considered all comments and suggestions from myself and the other reviewer. The inclusion of the new measures: total bird abundance, bird species richness and separate analyses of horse and cattle correlations to bird abundances have particularly improved the study.

Although it is good to avoid drawing conclusions of causal effects from correlative studies, I would say it is possible to speculate into what the results from a correlative study may suggest in a wider ecological context, while still making it clear that no causal effects can be confirmed. However, this is a matter of taste and the authors should be free in limiting their conclusions and speculations to what they feel is appropriate. The authors have given enough context to their study and enough motivation and explanation to their choices for the study to be accepted, in my opinion.